# Feasibility, Effectiveness and Safety of Elastomeric Pumps for Delivery of Antibiotics to Adult Hospital Inpatients—A Systematic Review

**DOI:** 10.3390/antibiotics12091351

**Published:** 2023-08-22

**Authors:** Joseph Spencer-Jones, Timothy Luxton, Stuart E. Bond, Jonathan Sandoe

**Affiliations:** 1Mid Yorkshire Teaching Trust, Wakefield WF1 4DG, UK; stuart.bond@nhs.net; 2School of Biomedical Sciences, University of Leeds, Leeds LS2 9JT, UK; bs13tl@leeds.ac.uk; 3School of Medicine, University of Leeds, Leeds LS2 9JT, UK; j.sandoe@leeds.ac.uk; 4Leeds Teaching Hospitals NHS Trust, Leeds LS9 7TF, UK

**Keywords:** elastomeric infusion pumps, antimicrobials, flucloxacillin, ceftazidime, intravenous, inpatient

## Abstract

Elastomeric infusion pumps (EMPs) have been implemented in many fields, including analgesia, chemotherapy and cardiology. Their application in antimicrobials is mainly limited to the outpatient setting, but with a need to optimise inpatient antimicrobial treatment, the use of EMPs presents a potential option. This review aimed to identify if the use of EMPs within an inpatient setting is feasible, effective and safe for antimicrobial use. Criteria for inclusion were human studies that involved the treatment of an infection with intravenous antimicrobial agents via an EMP. A search strategy was developed covering both the indexed and grey literature, with all study designs included. The review found 1 eligible study enrolling 6 patients. There was strong patient preference for EMPs (6/6), and daily tasks were easily completed whilst attached to the EMP. Nurses (5/5) also preffered the pumps, and the majority reported them as easy to use. The review has identified the need for further research in the area. Evidence for the use of EMPs to administer antibiotics in the inpatient setting is scarce, and more work is needed to understand the advantages to patients, to healthcare workers and from an antimicrobial stewardship perspective. Potential disadvantages that may put patients at risk also need investigating.

## 1. Introduction

Elastomeric infusion pumps (EMPs) are designed to safely and accurately deliver continuous infusions and have been successfully used to deliver infusions of analgesics, chemotherapy and cardiology treatments [1,2]. Also known as balloon pumps, EMPs deliver a continuous infusion by utilizing pressure created by an elastomeric balloon and, therefore, do not require a power source. Other advantages of EMPs are the ease of use, portability and fewer technical problems resulting in alarms [3].

EMPs do have disadvantages. The accuracy of the devices is low compared to electronic pumps, with a residual volume after infusion, meaning patients do not receive the full dose as prescribed as well as increased flow rates across all applications [4,5,6]. Zahnd et al. concluded that these characteristics are potentially dangerous for narrow therapeutic window drugs or those with long half-lives. However, in the application of chemotherapy, antimicrobials and analgesia, most drugs used have short half-lives, so many of these changes do not lead to accumulation [7]. EMPs also lack an alarm function whilst stability and environmental factors, such as temperature, will affect flow rate [5].

Use of EMPs for antibiotics is not new but is mainly confined to the outpatient setting, where studies have shown this method of administration to be safe and effective [8,9]. EMPs are included in the United Kingdom (UK) good practice guidelines for Out-patient Parenteral Antibiotic Therapy (OPAT) [10]. A recent review found that antibiotic EMPs optimized administration of time-dependent antibiotics and improved patients’ freedom [6]. Importantly, the continuous infusion of antibiotics requires less time to administer than multiple short infusions or bolus doses.

The need to optimize antimicrobial use is part of the UK 20-year strategy to tackle antimicrobial resistance (AMR), which is of vital importance given a recent systematic review estimated 1.27 million deaths in 2019 were attributable to AMR [11]. In the UK, one in three hospitalised patients are on an antibiotic at any given time [12], and each dose takes 20 minutes to prepare a standard infusion [13]. A recent survey by the Royal College of Nursing (RCN) found that 61% of nurses are too busy to provide the level of care they would like [14]. Workload is also a factor that contributes towards errors [15], so any intervention that can help to reduce this burden should be investigated.

In theory, EMPs may reduce the burden of AMR by allowing the administration of the narrowest spectrum agent possible by optimizing serum levels and reducing the number of missed doses. Although studies investigating efficacy of antimicrobials using continuous infusions have been inconclusive regarding mortality [16], continuous infusions of time-dependent beta-lactam antibiotics have shown to improve clinical cure and target plasma concentrations [17].

Whilst current studies present benefits in the outpatient setting with the need to optimize antimicrobials to prevent antimicrobial resistance, this review aims to identity if antibiotic delivery using EMPs in an inpatient setting is feasible, effective and safe.

## 2. Results

The search strategy identified 4403 articles, of which 1 met the inclusion criteria for a detailed review (Appendix B). The only paper that met the criteria was Cave et al. [18] (*n* = 6), an observational case series. Egerer et al. was initially thought to be suitable, but on further investigation, this was excluded as the infusion pumps used were battery powered [19]. Another paper, Hubert et al. [15], was excluded due to the lack of information about the number of participants treated as inpatients. 

### 2.1. Feasibility

Concerning the number of patients recruited to the studies, only 6 patients were recruited; however, Cave et al. did not include the number of patients approached/screened or a recruitment refusal rate. Cave et al. was limited to a single evaluation ward within one NHS hospital.

Cave et al. did not report any indications for the antibiotic used, but all patients were treated using the Baxter LV10 pump, administering flucloxacillin (8 g) over 24 h.

Cave et al. was the only study to report information about healthcare workers’ satisfaction, with all five nurses preferring an elastomeric pump and four out of five finding it very easy to administer.

The study reported patient experience with EMPs and the ease of day-to-day tasks with them attached. All six participants reported a 7 or above for ease of daily activity (0 was very difficult and 10 very easy) with participants experiencing short (intermittent) and continuous infusions via EMPs. All six patients preferred EMPs to the conventional six hourly infusions.

A daily cost comparison was also calculated, including staffing costs and time. For one pump, the daily cost was £ 74.95, compared with £ 32.00 for six hourly infusions.

### 2.2. Effectiveness and Safety

Cave et al. reported no information regarding clinical effectiveness or safety events during their analysis.

### 2.3. Risk of Bias Assessment

No randomized control trials were identified as being suitable. Cave et al. was assessed as having a critical risk of bias. This was an evaluation study focusing on qualitative feedback. The study provides a lack of information in general, including recruitment and refusal rate and dropouts. The nature of the study also means self-reporting is a key factor in the results (Appendix C).

## 3. Discussion

This systematic review is the first that has specifically focused on the inpatient use of antimicrobials via EMPs and confers that there is a distinct lack of evidence to guide and support the use of EMPs for inpatient antimicrobial use. It agrees with previous reviews investigating antimicrobial applications of EMPs that show that despite their use in a number of settings, there are very few studies on “real-life” work, with most published work relating to stability issues [6]. A recent systematic review of both chemical and physical stability of antimicrobials in EMPs found that more stability studies are required to ensure optimal patient outcomes. Another study concluded the same and acknowledged the need to study in real-life conditions [20]. Whilst a study investigating the drug degradation of amoxicillin in EMPs found that despite degradation exceeding 10%, as per recommendations, plasma concentrations remain sufficient to treat the infection [21].

Although small, the study by Cave et al. was fully inpatient based and included the potential time benefits of these EMPs within an inpatient setting. The study showed that EMPs can be used in an NHS setting, though there is not documented information about the supply chain or if these were pre-filled devices. Pre-filled devices can be obtained from homecare companies, but the compounding of these devices via pharmacy aseptic units is also common. There is also potential for nurses to compound EMPs; one OPAT study found this to be a safe and convenient alternative to pre-filled EMPs. This was in response to a limited supply [22]. All three options, commercially pre-filled, pharmacy preparation or nurse-compounded, will have different impacts on the cost, nursing time and stability of the supply chain and would need to be considered in terms of feasibility.

Interestingly, despite still being in the hospital, Cave et al. found that acceptability was high amongst the patients enrolled. This confers with other studies that investigated EMPs in OPAT, with patients preferring this method of delivery if threy required OPAT in the future [6,8,23,24].EMPs did not limit patients daily activity, something which is key in inpatient setting where immobility is known to be detrimental [25]. One study found that EMPs were preferred over electronically controlled pumps with the main reasons being: pump weight, impact on sleep and lack of technical problems [7,26]. Healthcare workers also reported that they found the EMPs easy to administer whilst it also saved them valuable time [27]. This is consistent with OPAT studies, a review of disposable infusion pumps found that patients felt more comfortable using the disposable device than an electronic pump, as did 80% of nurses [28].

However, despite the positives, there were still doubts amongst some patients about the devices [24]. Cave et al. did not report on device-related issues. This may be more relevant to outpatient studies and be less problematic in an inpatient setting where EMPs would be checked regularly by healthcare workers. However, based on previous reviews, emptying issues may be of concern. Despite this, they continue to be used in an OPAT setting. Whilst no outcome data regarding cure or failure of treatment were reported by Cave et al., previous OPAT studies have found EMPs to be clinically effective. However, the authors note the low levels of evidence supporting this [6,8,9].

Whist Cave et al. provides some cost information, including staffing time and drug costs, no formal health economic assessment is provided to help inform policy makers. They report that EMPs cost more than twice that of intermittent administration. This may vary for the type and frequency of the antibiotic being administered. Patients on EMPs may be able to be discharged sooner, and the benefits to patient mobility would need to considered.

The lack of inpatient studies is likely due to the advantages EMPs bring, such as allowing treatment at home, the ability to deliver cost efficiency savings due to reduced bed capacity and the improvement of the patient experience [23]. More research is required to understand the adverse effects, though other studies in the outpatient setting have found them to be safe [8]. However, due to the need for central venous access devices (CVAD), for administration of an EMP, there is a known risk for deep vein thromboses (DVTs) [29]. Cave et al. required a CVAD; in the case of a peripherally inserted central catheter (PICC), more understanding around the potential risks this may present is required.

### Limitations

An inclusive search strategy was developed and a large number of studies were screened, but we cannot exclude the possibility of missing studies, particularly conference abstracts. Cave et al. was found as part of conference proceedings and, therefore, provided limited information. Both authors were contacted for more information, but we were unsuccessful in obtaining a response. This impacted the interpretation of the results.

## 4. Materials and Methods

The review protocol is registered on PROSPERO (021 CRD42021286001).

### 4.1. Identification of Studies

A search strategy was developed with the aid of an experienced systematic searcher (CA) and agreed with the team.

A range of bibliographic sources were searched: AMED, CINAHL, Cochrane Library, EMBASE, MEDLINE and PubMed. Due to the lack of published data, a thorough search of the grey literature as well as hand searching was undertaken. OPENGREY, Web of Science, BIOSIS, Google and Google Scholar were the used databases.

In a change to the original search strategy, several other databases and grey literature sources were identified as well as manual searching of conference extracts, which can be found in the methods section. Key conferences were identified and hand searched, including Federation of Infection Societies, Healthcare Infection Society, British Society of Antimicrobial Chemotherapy and Infectious Diseases Society of America (IDSA). In addition, the reference lists of included studies were reviewed for potentially relevant papers. Searches were re-run in December 2022–February 2023, and additional papers were identified and screened, as over a year had passed since the initial search.

Appendix B contains the PRISMA flow diagram for studies. The search strategies used can be found in the Appendix A.

### 4.2. Criteria for Inclusion

The review considered human studies that involved the treatment of an infection with intravenous antimicrobial agents via an elastomeric pump. Studies of any research design were considered, and no language restrictions were applied.

Studies were excluded if they were only focusing on OPAT, Hospital in the home (HITH), ambulatory care or a setting that was not as a hospital inpatient. Studies were also excluded where it was not possible to distinguish where patients received treatment or the type of pump used. Further exclusion criteria included those studies that focused on drug stability alone and reported no patient related outcomes as well as those where the intravenous route was not used for drug delivery.

### 4.3. Selection of Studies

Titles and abstracts of all identified studies were uploaded into EndNote^®^ X9 and deduplication occurred, both using an automated approach as well as by hand. The remaining studies were transferred to Rayyan and blinded to avoid bias between reviewers. They were screened for eligibility by one reviewer (JSJ) and a random selection (10%) independently screened by a second reviewer (TL). Full text versions of papers not excluded at this stage were obtained for a detailed review by two reviewers (JSJ and TL). Any differences of opinion were reviewed by JS, who had overall validation.

### 4.4. Data Extraction

Data extraction was carried out by JSJ using an adapted Cochrane data collection from [30] on Microsoft Excel^®^. (Version 16.76) Information was collected on study aims and design, setting, duration, population, drug and delivery device and broad feasibility outcome measures (Figure A1). Individual authors were contacted for queries arising from data extraction.

### 4.5. Assessment of Bias

Studies were assessed by JSJ for bias using The Cochrane Risk of Bias assessment for RCTs, ROB2 with crossover [31,32] and the ROBINS-I tool used for non-RCTs [33]. The studies were then graded overall based on their risk of bias. Given the lack of randomized studies in the area, even those found to be at-risk were still included, but this was considered when interpretating these studies.

### 4.6. Data Synthesis

The characteristics of the included studies and findings relating to the outcomes specified were included in the data collection sheet (Appendix A).

The PRISMA [34] checklist and systematic review without meta-analysis (SWiM) were used to ensure that all relevant information was included.

## 5. Conclusions

There is very limited evidence available surrounding the feasibility of EMPS to administer antibiotics in the inpatient setting, though not impossible. A single small observational study provided no evidence to confirm the effectiveness or safety of EMPs to deliver antibiotics in an inpatient setting. Whilst evidence can be extrapolated from the outpatient setting, the lack of robust evidence in this area is also limited despite the extensive use of EMPs for OPAT.

Issues with residual volume, dosing and degradation of drug above the recommended limits, present real questions, especially in the context of antimicrobial resistance. However, despite the issues “clinical cure” and successful outcomes are still reported. Understanding this is vitally important given the need to optimize antimicrobial dosing in the fight against resistance.

More work is needed, both in the inpatient and the outpatient setting, to understand the feasibility, safety and effectiveness of these devices whilst also exploring the potential advantages to patients, healthcare workers and for antimicrobial stewardship perspective.

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
