# Peer review of "Feasibility, Effectiveness and Safety of Elastomeric Pumps for Delivery of Antibiotics to Adult Hospital Inpatients—A Systematic Review"

_antibiotics, 2023, doi:10.3390/antibiotics12091351_

Round 1

Reviewer 1 Report

accept after minor revision

accept after minor revision

Author Response

Added further references as per other review comments.

Reviewer 2 Report

The review is of great importance. But unfortunately less research has been carried out. Writing a review on such limited data is questionable. There are few suggestions for the authors;

1. Give your opinion  the use of EMPs for antibiotics in inpatient vs the use of EMPs for chemotherapy. 

2. What is the possible reason that the health care provider opted for EMPs in outpatients antimicrobial use. Why inpatients are not considered much. 

2. The conclusion has same sentence as in abstract. The conclusion at the end of the article should be relatively more detailed, highlighting the antibiotic stewardship.  

3. There are few references which are repeated again and again in the article e.g. reference [3].

Author Response

  1. Issues seen with flow rates / residual volume have been seen in chemotherapy EMPs. But it is thought that this will only affect those drugs with narrow therapeutic windows or with long half lives. Which would lead to accumulation and toxicity. Chemotherapy and antimicrobials are usually those with short half lives. In both instances EMPs have been found to be effective.
  2. The main reason for lack of use in inpatients is likely related to cost savings and the ability to discharge. Once an EMP is fitted the patient can be considered for OPAT. However in some circumstances such as endocarditis this may not be possible due to monitoring required. The patient may also not be eligible, for example as they are an intravenous drug user. 
  3. I have adde some further information to the abstract including the fact that some degredation above the recommended limits was seen in one study with amoxicillin. Despite this plasma levels were in range. This potentially suggests dosing could be lowered, playing a part in the fight against resistance. 
  4. I have added some further references found from the systematic review, mainly relating to EMPs within OPAT. 

Reviewer 3 Report

It is very difficult to comment a review that ultimately consider only one study with six patients. I don't even know if it has any value as a review.

I suggest to the Autors either to wait for new research in this field or to be less restrictive in the their Inclusion criteria, so it doesn't make much sense, in my opinion.

Author Response

Thank you for the comments. 

Whilst there is certainly a distinct lack of evidence for use in the inpatient setting, this is also the case in the outpatient setting. 

Issues with residual volume and potential for reduced doses, unclear efficacy from a clinical cure perspective and the need to optimise dosing given the significant threat of antimicrobial resistance are all unanswered questions for future research. 

This review hopefully should highlight the fact that despite extensive use of these devices, there is limited evidence of real world data, in an inpatient setting, but also outpatients. Most evidence is based on stability, in the lab or extrapolated from intermittent dosing.

Round 2

Reviewer 2 Report

The revised manuscript is much improved.